Anti-apoptotic properties of carbon monoxide in porcine oocyte during in vitro aging

Němeček David nemecekd@af.czu.cz
Dvořáková Markéta
Heroutová Ivona
Chmelíková Eva
Sedmíková Markéta
Department of Veterinary Sciences, Czech University of Life Sciences , Prague , Czech Republic
Wessel Gary
Electronic publication date: 2017 Oct 6
Publication date: 2017
Volume: 5
Electronic Location ID: e3876
Received 2017 Apr 4; Accepted 2017 Sep 9
Copyright: ©2017 Němeček et al.
Copyright year: 2017
Copyright holder: Němeček et al.
License: This is an open access article distributed under the terms of the Creative Commons Attribution License, which permits unrestricted use, distribution, reproduction and adaptation in any medium and for any purpose provided that it is properly attributed. For attribution, the original author(s), title, publication source (PeerJ) and either DOI or URL of the article must be cited.
License URL: https://creativecommons.org/licenses/by/4.0/

Keywords: Carbon monoxide, Heme oxygenase, Oocyte, Pigs, Aging, Antiapoptotic, Caspase-3

Funding: Ministry of Agriculture of the Czech Republic NAZV QJ1510138 Internal Grant Agency of the Czech University of Life Sciences Prague (CIGA) CZU20152022 CZU20142049 This work was supported by the Ministry of Agriculture of the Czech Republic (NAZV–Project No. QJ1510138) and by Internal Grant Agency of the Czech University of Life Sciences Prague (CIGA) (Projects No. CZU20152022 and CZU20142049). The funders had no role in study design, data collection and analysis, decision to publish, or preparation of the manuscript.

==============================
If fertilization of matured oocyte does not occur, unfertilized oocyte undergoes aging, resulting in a time-dependent reduction of the oocyte’s quality. The aging of porcine oocytes can lead to apoptosis. Carbon monoxide (CO), a signal molecule produced by the heme oxygenase (HO), possesses cytoprotective and anti-apoptotic effects that have been described in somatic cells. However, the effects of CO in oocytes have yet to be investigated. By immunocytochemistry method we detected that both isoforms of heme oxygenase (HO-1 and HO-2) are present in the porcine oocytes. Based on the morphological signs of oocyte aging, it was found that the inhibition of both HO isoforms by Zn-protoporphyrin IX (Zn-PP IX) leads to an increase in the number of apoptotic oocytes and decrease in the number of intact oocytes during aging. Contrarily, the presence of CO donors (CORM-2 or CORM-A1) significantly decrease the number of apoptotic oocytes while increasing the number of intact oocytes. We also determined that CO donors significantly decrease the caspase-3 (CAS-3) activity. Our results suggest that HO/CO contributes to the sustaining viability through regulation of apoptosis during in vitro aging of porcine oocytes.

Introduction

In most mammals, a mature oocyte is in the stage of the metaphase of the second meiotic division, when it awaits fertilization. If the fertilization does not occur within the time referred to as a ‘temporal window for optimal fertilization’ (Fissore et al., 2002; Goud et al., 2005), a time-dependent decrease in oocyte quality takes place, which is also referred to as post-ovulatory oocyte aging (Fissore et al., 2002; Miao et al., 2009; Lord & Aitken, 2013). In pigs, as well as in other mammals, the aging process takes place in both in vivo and in vitro conditions (Petrová et al., 2004; Petrová et al., 2009). Oocyte aging is one of the factors limiting various assisted reproductive technologies (ART) outcome in several mammalian species (Miao et al., 2009). Many changes of cellular functions occur during oocyte aging, as well as morphological changes of the cytoskeleton and cellular organelles. The negative effects of aging include premature exocytosis of cortical granules (Szollosi, 1971), structural changes of the zona pellucida (Xu et al., 1997), the decrease of the fertilizing capability (Lanman, 1968), increase of polyspermy (Badenas et al., 1989), parthenogenesis (Blandau, 1952) and chromosomal aberrations (Szollosi, 1971).

Several mechanisms contribute to the formation of negative effects of the oocyte aging. Aging process leads to the progressive increase in ROS production and the concomitant depletion of antioxidant protection and as a consequence the post ovulatory aged oocyte experiences a state of oxidative stress (Lord & Aitken, 2013). This relates to the disruption in functions of the mitochondria and Ca2+ signaling (Liu, Trimarchi & Keefe, 2000; Lord & Aitken, 2013). Changes in the activity of the M-phase promotion factor (MPF) and the mitogen-activated protein kinase (MAPK) that maintain the meiotic arrest in metaphase II also occurs during aging (Kikuchi et al., 1995; Miao et al., 2009; Jiang et al., 2011). The decrease of the MPF activity causes parthenogenetic activation in the aged oocytes and consequently cellular death. The increased MAPK activity also contributes to the triggering of cellular death (Sadler et al., 2004; Ješeta et al., 2008; Miao et al., 2009). Aging process finally leads to lytic or, more often, apoptotic cell death of aged oocytes (Fissore et al., 2002; Miao et al., 2009; Petrová et al., 2009; Lord & Aitken, 2013).

Programmed cell death is characterized by the activation of caspases (aspartate-specific cysteine proteases) that are activated upon the receipt of either an extrinsic or intrinsic death signal. Both signals induce the execution phase of the apoptotic pathway characterized by the activation of executioner caspases that subsequently activate cytoplasmic endonucleases and proteases. Their activation leads to characteristic morphological and biochemical changes observed during apoptosis (Salvesen & Dixit, 1997; Slee, Adrain & Martin, 2001; Taylor, Cullen & Martin, 2008). Caspase-3 (CAS-3) is one of the most important executioner caspases. The CAS-3 activity is often used as a marker of apoptotic cell death, regardless of whether the apoptosis was triggered through an extrinsic or intrinsic pathway (Elmore, 2007). Similarly, as in somatic cells, also in aged oocytes is CAS-3 activated during apoptotic cell death (Zhu et al., 2015; Zhu et al., 2016).

Carbon monoxide (CO), endogenously produced by heme oxygenase (HO) or exogenously delivered by CO gas or CO-releasing molecules (CORMs) (Motterlini et al., 2003) is one of the known factors that can modulate apoptotic pathway in various types of somatic cells (Brouard et al., 2000; Petrache et al., 2000; Wu & Wang, 2005; Ryter, Alam & Choi, 2006; Kim et al., 2011), but the effect of CO in oocytes is unknown. HO enzyme catalyzes oxidative cleavage of heme producing ferrous iron, biliverdin-IXα and CO (Tenhunen, Marver & Schmid, 1968; Tenhunen, Marver & Schmid, 1969). HO exists in two active isoforms, HO-1 and HO-2. HO-1 is an inducible isoform activated by different kinds of stresses (e.g., oxidative stress) (Biswas et al., 2014; Ryter & Choi, 2016), while the constitutive isoform HO-2 is responsible for the HO basal activity (Turkseven et al., 2007; Muñoz Sánchez & Chánez-Cárdenas, 2014).

CO influences a variety of signalling pathways and generally has cytoprotective, anti-apoptotic and anti-inflammatory properties (Motterlini & Otterbein, 2010). In murine endothelial cells CO suppresses apoptosis through activation of the p38 MAPK (Brouard et al., 2000; Brouard et al., 2002) and in rat endothelial cells CO prevents the initiation of apoptosis through increasing the expression of anti-apoptotic factor Bcl-2 (Zhang et al., 2003a) and decreasing the expression/activation of pro-apoptotic factors Bid and Bax (Zhang et al., 2003a; Zhang et al., 2003b; Wang et al., 2007a). It further prevents the release of cytochrome c from mitochondrial matrix (Zhang et al., 2003a; Wang et al., 2007a). Overall, CO decreases caspases activation (CAS-3, CAS-8, CAS-9), which was proven in endothelial cells of mice and rats (Zhang et al., 2003a; Zhang et al., 2003b; Wang et al., 2007a; Wang et al., 2011), in murine astrocytes (Almeida et al., 2012), rat ganglion cells (Schallner et al., 2012) and porcine lung tissue (Goebel et al., 2008).

The anti-apoptotic effect of the HO/CO system was investigated in somatic cells, but in the case of oocytes the effect of CO is so far unknown. While it was proven that HO-1 deficiency in mice causes lower fertilization capability of oocytes (Zenclussen et al., 2012), knowledge of the HO/CO significance in oocytes is still insufficient. We assumed that, similarly as in somatic cells, CO could also prevent the apoptotic pathway in oocytes. We hypothesized that CO could improve the viability of porcine oocytes. The aim of our work is to identify the effect of CO on the course of the porcine oocyte aging.

Materials and Methods

Collection, cultivation and in vitro aging of porcine oocytes

Porcine ovaries were obtained from local slaughterhouses from gilts during an unknown stage of the oestrous cycle. Porcine oocytes were obtained from the ovaries through aspiration of the follicular fluid from follicles (2–5 mm). Only oocytes with intact cytoplasm and compact cumuli were chosen for the experiments. Oocytes were cultivated in the modified M199 medium (Gibco-BRL, Life Technologies, Paisley, Scotland) containing sodium bicarbonate (32.5 mM), calcium L-lactate (2.75 mM), gentamicin (0.025 mg/ml), HEPES (6.3 mM), 13.5 IU eCG: 6.6 IU hCG/ml (P.G.600; Intervet, Boxmeer, Holland) and 10% (v/v) fetal calf serum (GibcoBRL). Oocytes were cultivated for 48 h up to the metaphase stage of the second meiotic division (MII) in 4-well Petri dishes (Nunc, Fisher Scientific, Waltham, MA, USA; 1 ml of cultivation medium; 39 °C; 5,0 % CO2). Meiotically matured oocytes were denuded and used for in vitro aging (24, 48 and 72 h) under the same conditions in the cultivation medium without P.G. 600.

Evaluation of in vitro aged porcine oocytes

Upon completion of the in vitro aging, the oocytes were fixed in a mixture of ethanol and acetic acid (3:1, w/v, 48 h). Fixed oocytes were stained using 1.0% (w/v) orcein. The oocytes were classified into four groups according to the morphological signs of aging: intact oocytes (oocytes in the stages: metaphase II, anaphase II or telophase II), parthenogenetically activated oocytes (embryos and oocytes containing pronuclei), apoptotic oocytes (called fragmented oocytes; oocytes containing apoptotic vesicles under the zona pellucida) and lysed oocytes (oocytes with loss of integrity and rupture of cytoplasmic membrane) (Petrová et al., 2004).

Immunocytochemical detection of heme oxygenase and cleaved caspase-3

Upon completion of the oocyte cultivation period, zona pellucida was removed (0.1% pronase). Subsequently, the oocytes were fixed in 2.5% (w/v) paraformaldehyde in Phosphate-buffered saline (PBS). The oocyte membrane was permeabilized by 0.5% (w/v) Triton X in PBS with 0.01% (w/v) BSA. After rinsing in PBS with 0.1% (w/v) Tween 20 followed incubation with the primary antibody anti-heme oxygenase 1 or anti-heme oxygenase 2 (Abnova Corporation, Taipei, Taiwan; 1:200) or anti-cleaved caspase-3 (Asp175) (Cell Signaling Technology, Danvers, USA; 1:400). The incubation period took over 14-16 h in moisture at a temperature of 4 °C in 0.1 % (w/v) BSA and 0.01 % (w/v) Tween 20 in PBS. After incubation, the oocytes were rinsed with 0.1% (w/v) Tween 20 in PBS and cultivated with secondary anti-mouse IgG antibody conjugated with fluorescein-5-isothiocyanate (FITC, Sigma–Aldrich Gmbh, Munich, Germany; 1:100) or anti-rabbit IgG conjugated with FITC (ThermoFisher Scientific, Rockford, USA; 1:500). The incubation with the secondary antibody took place at laboratory temperature in 0.1% (w/v) BSA and 0.01% (w/v) Tween 20 in PBS over the course of one hour in darkness. After incubation, the oocytes were rinsed in 0.1% (w/v) Tween 20 in PBS.

The chromatin was stained using 4′,6-diamidino-2-phenylindole (DAPI, Sigma–Aldrich Gmbh, Munich, Germany). To exclude non-specific secondary antibody binding the oocytes in the control group were treated in the same way as the experimental group, except that primary antibody incubation was not performed. Oocytes were mounted on slides and images were acquired using confocal scanning microscope (Zeiss, Germany). The images were analyzed in NIS Elements AR Software (NIKON, Japan). The data was expressed as mean signal intensity of the FITC fluorescence, reduced by a basal signal intensity of appropriate negative control. Each experiment was repeated at least three times at a minimal amount of 15 oocytes in each experimental group.

Western blot

Western blot was performed for validation of the antibodies specificity in accordance to Tůmová et al. (2013). Briefly, MII oocytes were lysed and separated by SDS-PAGE. After blotting the membranes were incubated with primary antibodies—anti-heme oxygenase-1, anti-heme oxygenase-2 (Abnova Corporation, Taipei, Taiwan; 1:1,000) or cleaved caspase-3 (Asp175) (Cell Signaling Technology, Danvers, MA, USA; 1:1,000) and then with secondary antibodies—Mouse IgG (Amersham GE Healthcare, Life Sciences, Little Chalfont, Buckinghamshire, United Kingdom; 1:30,000) or rabbit IgG (Amersham GE Healthcare, Life Sciences, United Kingdom; 1:120,000). Proteins were detected by ECL Advanced Western blotting detection kit (Amersham GE Healthcare, Life Sciences, United Kingdom). Each western blot was repeated at least three times with a minimal amount of 200 oocytes.

In vitro cultivation of aging oocytes with exogenous CO donors or heme oxygenase inhibitor

Oocytes were cultivated in a cultivation media with the CO donors: CORM-2 (tricarbonyl dichlororuthenium (II) dimer; Sigma–Aldrich Gmbh, Munich, Germany) at concentrations of 5, 25, 50 and 100 µM, dissolved in dimethyl sulfoxide (DMSO) or CORM-A1 (sodium boranocarbonate; Sigma–Aldrich Gmbh, Munich, Germany) at concentrations of 25, 50 and 100 µM, dissolved in H2O. CORM-2 and CORM-A1 both have distinct rate of CO release, wherein CORM-2 is a fast CO releaser, whereas CORM-A1 shows a slow and gradual CO release (Motterlini et al., 2002; Motterlini, Mann & Foresti, 2005). In the case of HO inhibition, the oocytes were cultivated in a cultivation medium with HO inhibitor Zn-protoporphyrin IX (Zn-PP IX; Sigma–Aldrich Gmbh, Munich, Germany), at concentrations of 2.5, 5 and 25 µM dissolved in DMSO. In order to eliminate the effect of DMSO on the aging of porcine oocytes, a control group was incubated in a cultivation medium containing only DMSO. In order to confirm the effect of CO, oocytes were cultivated in cultivation medium with ruthenium (III) chloride (inactive CORM2; iCORM-2; Sigma–Aldrich Gmbh, Munich, Germany) or inactive CORM-A1 (iCORM-A1). iCORM-A1 consisted of CORM-A1 prepared in 0.1 M HCl bubbled with N2 gas for 10 min to dissipate all of the CO and then the pH of the solution was adjusted to 7.4. The effect of the iCORMs at comparable concentrations was not significant compared to oocytes cultivated in pure cultivation medium (Table S1). Upon completion of the given cultivation period, the oocytes were morphologically evaluated or prepared for immunocytochemical technique. Each experiment was repeated at least three times and at least 80 oocytes were used for each experimental group.

Statistical analysis

Data from all experiments were subjected to statistical analysis. All experiments were repeated at least three times. The SAS 9.0 software (SAS Institute Inc., Cary, NC, USA) was used for statistical analysis. Significant differences between groups were determined using analysis of variance (ANOVA) followed by Scheffe’s method. A p-value of less than 0.05 was considered significant. Significant differences among different groups of oocytes are indicated by different symbols.

Results

Heme oxygenase isoforms are present in porcine oocyte during in vitro aging

The objective of the experiment was to localize HO-1 and HO-2 in meiotically mature porcine oocytes (MII) and oocytes exposed to in vitro aging for the period of 24, 48 and 72 h. HO isoforms are present in porcine oocyte (Figs. 1 and 2), and their expression gradually increase during in vitro aging. In case of HO-1, the signal was predominant in the nucleus/perichromosomal area (Fig. 2). On the contrary, the HO-2 signal was mainly observed in the oocyte cytoplasm (Fig. 3).

Figure 1 Immunoblotting of HO isoforms (HO-2, MW∼35,7 kDa; HO-1, MW∼33/28 kDa) in porcine oocytes.

Oocytes were matured to the second metaphase stage (MII stage). Proteins were separated by SDS-PAGE, transferred to a nitrocellulose membrane and then incubated with HO specific antibodies (anti-HO1, anti-HO2; both 1:1,000). One sample contained proteins from 200 oocytes. HO1 and HO2 and their truncated forms were detected by specific antibodies. The arrows indicate bands corresponding to the molecular weight of the HO protein.

Figure 2 Localization of HO-1 in meiotically matured porcine oocytes (MII) (A) and in oocytes exposed to in vitro aging for 24 (B), 48 (C) and 72 (D) hours.

HO-1 is shown in green (FITC), chromatin is shown in blue (DAPI), magnified 400×.

Figure 3 Localization of HO-2 in meiotically mature porcine oocytes (MII) (A) and in oocytes exposed to in vitro aging for 24 (B), 48 (C) and 72 (D) hours.

HO-2 is shown in green (FITC), chromatin is shown in blue (DAPI), magnified 400×.

During the in vitro aging, the expression of both isoforms increased primarily in the oocyte cytoplasm. In comparison with meiotically matured oocytes, in oocytes aged 24 h the expression of HO-1 increased by 2.2 ± 0.2 times, aged 48 h increased by 3.5 ± 0.3 times and aged 72 h increased by 6.7 ± 1.3 times (Fig. 4A). In the case of HO-2, the expression in oocytes aged 24 h increased by 1.3 ± 0.1 times, aged 48 h increased by 1.7 ± 0.14 times, and in oocytes aged 72 h increased by 3.1 ± 0.6 times (Fig. 4B).

Figure 4 Expression of HO-1 (A) and HO-2 (B) in meiotically matured (MII) porcine oocytes and oocytes exposed to in vitro aging for 24, 48 and 72 h.

The level of expression of HO-1 and HO-2 was determined as the mean intensity of the given isoform’s signal in porcine oocytes and related to the average signal intensity of the given HO isoform in meiotically matured (MII) oocytes. Bars show the mean ± SEM. *# + indicates significant differences in the HO-1 and HO-2 signal intensity (P < 0.05).

The inhibition of heme oxygenase increases the ratio of apoptotic porcine oocytes during the in vitro aging

This experiment focused on the effect of HO inhibition by HO inhibitor Zn-protoporphyrin IX (Zn-PP IX) on in vitro aging of porcine oocytes. After 24 h of in vitro aging, the effect was significant only when the highest concentration of Zn-PP IX (25 µM) was used, where the amount of intact oocytes in comparison to the control group decreased by 10.6% (94.0 ± 1.5 vs. 83.4 ± 2.1% for control and HO inhibitor, respectively) (Fig. 5A). In other concentrations of Zn-PP IX, the effect on the morphological signs of aging was not significant.

Figure 5 The effect of the HO inhibitor zinc protoporphyrin IX (Zn-PP IX) on the porcine oocytes during in vitro aging for 24 (A), 48 (B) and 72 (C) hours.

Control group was cultivated in a medium containing DMSO. The experimental group was cultivated in a medium containing Zn-PP IX at the concentrations of 2.5, 5 and 25 µM. Data is expressed in a relative manner. The stages of in vitro aging were morphologically evaluated as: intact (MII), lytic (L), parthenogenetically activated (PA) and apoptotic (A). * indicates significant difference in the ratio of intact (MII) oocytes between different concentrations of inhibitor and control group (P < 0.05). # indicates significant difference in ratio of apoptotic (A) oocytes between different concentrations of inhibitor and control group (P < 0.05). + indicates significant difference in ratio of lytic (L) oocytes between different concentrations of inhibitor and control group. No significant difference was found in the ration of parthenogenetically activated (PA) oocytes between different concentrations of inhibitor and control group.

After 48 h of in vitro aging in the presence of Zn-PP IX, the effect was significant in all concentrations, causing decreases in the ratio of intact oocytes and increases in the ratio of apoptotic oocytes. In oocytes treated with Zn-PP IX, the ratio of intact oocytes decreased by 11.1–12.9% (67.1 ± 1.6 vs. 54.2 ± 2.1–56.0 ± 1.7% for control and HO inhibitor, respectively) whiles the ratio of apoptotic oocytes increased by 5.6–9.4% (21.5 ± 2.3 vs. 27.0 ± 1.1–30.8 ± 2.4% for control and HO inhibitor, respectively). Moreover, the highest concentration of inhibitor (25 µM) also increased the ratio of lytic oocytes by 8.3% (1.1 ± 1.5 vs. 9.4 ± 1.6% for control and HO inhibitor, respectively). The effects of different concentrations of inhibitor did not differ significantly (Fig. 5B).

The most obvious manifestation of negative signs of aging was observed in oocytes aged 72 h, wherein the ratio of apoptotic oocytes reached 60.4 ± 3.7–67.5 ± 3.4% in all groups (control and experimental). However, the effect of the HO inhibitor was not significant (Fig. 5C).

CO donors suppress apoptosis in porcine oocytes during the in vitro aging

The objective of this experiment was to identify the effect of CO on the course of in vitro aging of porcine oocytes evaluated based on morphological signs of aging. As the CO source, we used two CO donors (CORM-2 or CORM-A1) each with different CO release kinetic. The control group of oocytes was cultivated in the presence of inactive compounds (iCORM-2 or iCORM-A1) to eliminate its effect. It was found that CO delivered by CORMs causing decreases in the ratio of apoptotic oocytes and increases in the ratio of intact oocytes.

The effect of the CO donors on in vitro aging became apparent after 48 h of in vitro aging. The effect of CORM-A1 was significant at the concentrations 25 and 50 µM, which decreased the ratio of apoptotic oocytes by 10.2–14.4% (30.3 ± 4.3 vs. 15.6 ± 5.8–19.8 ± 4.6 for control and CORM-A1, respectively) and simultaneously increased the ration of intact oocytes by 12.6–16.0% (59.8 ± 3.6 vs. 72.3 ± 3.8–75.7 ± 3.7 for control and CORM-A1, respectively) (Figs. 6A and 6B). The difference between control group and group cultivated in the CORM-A1 at the concentration 100 µM was not significant. The effect of CORM-2 was significant in all concentrations. CORM-2 decreased the ratio of apoptotic oocytes by 6.7–9.9% (21.5 ± 1.3 vs. 11.5 ± 1.5–14.8 ± 0.3% for control and CORM-2, respectively), while simultaneously increased the ratio of intact oocytes by 8.6–13.5% (67.1 ± 1.6 vs. 75.7 ± 1.4–80.6 ± 1.4% for control and CORM-2, respectively). The effects of different concentrations of CORM-2 did not differ significantly (Figs. 7A and 7B).

Figure 6 The effect of the CO donor CORM-A1 on the porcine oocytes during in vitro aging for 24 (A), 48 (B) and 72 (C) hours.

The control group was cultivated in a medium containing an inactive form of CO donor (iCORM-A1; 100 µM). The experimental group was cultivated in a medium containing CORM-A1 at the concentrations of 25, 50 and 100 µM. The data is expressed in a relative manner. Stages of in vitro aging were determined morphologically as: intact (MII), lytic (L), parthenogenetically activated (PA) and apoptotic (A). * indicates significant difference in the ratio of intact (MII) oocytes between different concentrations of CORM-A1 and control group (P < 0.05). # indicates significant difference in the ratio of apoptotic (A) oocytes between different concentrations of CORM-A1 and control group (P < 0.05). No significant difference was found in the ration of parthenogenetically activated (PA) and lytic (L) oocytes between different concentrations of CORM-A1 and control group.

Figure 7 The effect of the CO donor CORM-2 on the porcine oocytes during in vitro aging for 24 (A), 48 (B) and 72 (C) hours.

The control group (C) was cultivated in a medium containing an inactive form of CO donor (iCORM-2; 100 µM). The experimental group was cultivated in a medium containing CORM-2 at the concentrations of 5, 25, 50 and 100 µM. The data is expressed in a relative manner. Stages of in vitro aging were determined morphologically as: intact (MII), lytic (L), parthenogenetically activated (PA) and apoptotic (A). * indicates significant difference in the ratio of intact (MII) oocytes between different concentrations of CORM-A1 and control group (P < 0.05). # indicates significant difference in the ratio of apoptotic (A) oocytes between different concentrations of CORM-A1 and control group (P < 0.05). No significant difference was found in the ration of parthenogenetically activated (PA) and lytic (L) oocytes between different concentrations of CORM-2 and control group.

Both CO donors significantly suppressed apoptosis even in oocytes aged 72 h. The effect of CORM-A1 was significant at the concentrations 50 and 100 µM, which decreased the ratio of apoptotic oocytes by 12.3–14.3% (59.5 ± 3.6 vs. 45.2 ± 7.3–47.2 ± 3.2 for control and CORM-A1, respectively) and simultaneously increased the ratio of intact oocytes by 11.2–12.5% (28.8 ± 3.7 vs. 40.0 ± 7.3–41.3 ± 6.0 for control and CORM-A1, respectively) (Fig. 6C). The difference between control group and group cultivated in the CORM-A1 concentration 25 µM was not significant. As with oocytes aged 48 h, also in oocytes aged 72 h the effect of CORM-2 was significant in all concentrations. CORM-2 decreased the ratio of apoptotic oocytes by 8.9–17.4% (60.4 ± 2.7 vs. 43.0 ± 3.6–51.5 ± 1.5% for control and CORM-2, respectively) while increased the ratio of intact oocytes by 12.6–21.6% (17.5 ± 1.7 vs. 30.1 ± 2.4–39.1 ± 2.2% for control and CORM-2, respectively). The effects of different concentrations of CORM-2 did not differ significantly (Fig. 7C).

CO donors decrease the activity of caspase-3 in porcine oocytes during the in vitro aging

This experiment focused on the effect of CO donors CORM-2 or CORM-A1 on the fluorescence intensity of activated CAS-3 (cleaved CAS-3; cCAS-3) as a marker of apoptosis. It was found that the CO donors decrease the fluorescence intensity of cCAS-3 during in vitro aging (Figs. 8–10).

Figure 8 Typical expression pattern of activated CAS-3 (cleaved CAS-3) in porcine oocytes during in vitro aging for 24 (A), 48 (B) and 72 (C) hours in the presence of CO donor CORM-A1 at the concentrations of 25 and 50 µM.

The control group represents oocytes cultivated with an inactive form of the CO donor (iCORM-A1). The experimental group represents oocytes cultivated with CORM-A1. The negative control group (NC) represents oocytes treated only with secondary antibody. Cleaved CAS-3 is shown in green (FITC), chromatin is shown in blue (DAPI), magnified 400×.

Figure 9 Typical expression pattern of activated CAS-3 (cleaved CAS-3) in porcine oocytes during in vitro aging for 24 (A), 48 (B) and 72 (C) hours in the presence of CO donor CORM-2.

The control group represents oocytes cultivated with an inactive form of the CO donor (iCORM-2). The experimental group represents oocytes cultivated with CORM-2. The negative control group (NC) represents oocytes treated only with secondary antibody. Cleaved CAS-3 is shown in green (FITC), chromatin is shown in blue (DAPI), magnified 400×.

Figure 10 Immunoblotting of cleaved caspase 3 (cCAS-3, MW∼17/19 kDa) in porcine oocytes.

Oocytes were meiotically matured in the second metaphase (MII stage). Proteins were separated on SDS-PAGE, transferred to a nitrocellulose membrane and then incubated with CAS-3 specific antibodies (anti-cleaved caspase-3 (Asp175). One sample contained proteins from 400 oocytes.

The effect of both CO donors on the activity of CAS-3 was significant in all groups, i.e., in oocytes aged 24, 48 and 72 h. In the case of CORM-2, the effect of different concentrations did not differ significantly, whereas the effect of CORM-A1 differs depending to the concentration. We observed the most significant effect of both CO donors in oocytes aged 24 h, wherein CORM-A1 causing decreases of cCAS- 3 fluorescence intensity by 40.9–48.5% (100.0 ± 6.0 vs. 51.6 ± 6.2–59.2 ± 5.9% for control and CORM-A1 at the concentrations 25 and 50 µM, respectively). The effect of CORM-A1 at the concentration 100 µM was less significant and decreased cCAS-3 fluorescence intensity by 16.5% (100.0 ± 6.0 vs. 83.5 ± 5.2% for control and CORM-A1 at the concentrations 100 µM, respectively). In the case of CORM-2, the cCAS-3 fluorescence intensity decreased by 55.6–63.7% (100.0 ± 8.5 vs. 36.3 ± 4.5–44.4 ± 2.7% for control and CORM-2, respectively).

After 48 and 72 h of in vitro aging, the cCAS-3 fluorescence intensity was significantly lower in comparison with oocytes aged 24 h, however, the cultivation of oocytes with the CO donors also lead to the significant decrease of cCAS-3 fluorescence intensity. In group of oocytes aged 48 h with CORM-A1 the cCAS-3 fluorescence intensity decreased by 16.7–23.5% (65.6 ± 10.3 vs. 42.1 ± 4.5–48.9 ± 4.9 for control and CORM-A1 at the concentrations 25 and 50 µM, respectively). The effect of CORM-A1 at the concentration 100 µM was not significant (Fig. 11). In the case of CORM-2 the cCAS-3 fluorescence intensity decreased by 13.1–22.4% (35.3 ± 2.9 vs. 13.1 ± 1.8–22.4 ± 1.8 for control and CORM-2, respectively) in oocytes aged 48 h (Fig. 12). In group of oocytes aged 72 h with CORM-A1 the cCAS-3 fluorescence intensity decreased by 13.4-22.9% (69.0 ± 7.6 vs. 46.1 ± 8.7–55.6 ± 5.2 for control and CORM-A1 at the concentrations 25 and 50 µM, respectively). The effect of CORM-A1 at the concentration 100 µM was not significant. In the case of CORM-2 the cCAS-3 fluorescence intensity decreased by 10.8–14.1% (24.9 ± 5.2 vs. 10.8 ± 3.8–14.1 ± 2.0 for control and CORM-2, respectively) in oocytes aged 72 h.

Figure 11 The effect of the CO donor CORM-A1 on the expression of activated CAS-3 during in vitro aging of porcine oocytes for 24, 48 and 72 h.

The data of expression of the activated CAS-3 are related to the data of expression of the activated CAS-3 in oocytes in the control group, exposed to in vitro aging for 24 h. The control group represents oocytes cultivated in a medium containing an inactive form of the CO donor (iCORM-A1; 100 µM). The experimental group was cultivated in a medium containing CORM-A1 at the concentrations of 25, 50 and 100 µM. Bars show mean ± SEM. *# indicate significant differences in the level of expression of the activated CAS-3 between different concentrations of CORM-A1 or control group for each aging time separately (P < 0.05).

Figure 12 The effect of the CO donor CORM-2 on the expression of activated CAS-3 during in vitro aging of porcine oocytes for 24, 48 and 72 h.

The data of expression of the activated CAS-3 are related to the data of expression of the activated CAS-3 in oocytes in the control group exposed to in vitro aging for 24 h. The control group represents oocytes cultivated in a medium containing an inactive form of the CO donor (iCORM-2; 100 µM). The experimental group was cultivated in a medium containing CORM-2 at the concentrations of 25, 50 and 100 µM. Bars show mean ± SEM. * indicates significant differences in the level of expression of the activated CAS-3 between different concentrations of CORM-2 and control group for each aging time separately (P < 0.05).

Discussion

Post-ovulatory oocyte aging causes progressive decrease in oocyte quality and ultimately leads to cell death. Heme oxygenase (HO) is essential enzyme which cytoprotective properties being caused by the production of an important signalling molecule, carbon monoxide (CO) (Wu & Wang, 2005). Together with hydrogen sulfide (H2S) and nitric oxide (NO), CO belongs to gasotransmitters, which are a subfamily of endogenous molecules of gases or gaseous signalling molecules. The role of H2S and NO as signal molecules in porcine oocytes was identified (Bu et al., 2003; Goud et al., 2005; Nevoral et al., 2014; Krejčová et al., 2015), but information regarding the function of the HO/CO system is yet to be unavailable. Our work proves the presence of both HO isoforms in porcine oocytes and characterizes the effect of the HO inhibitor and CO donors on the course of in vitro aging of porcine oocytes.

We have shown that both HO isoforms (HO-1 and HO-2) are localized in meiotically matured and aged porcine oocytes and that over the course of in vitro aging, the expression of both HO isoforms increase. The HO-1 is a member of the superfamily of stress proteins, the expression of which increase in reaction to a wide spectrum of inducers (for example oxidative stress) (Dennery, 2000; Ryter, Alam & Choi, 2006). The increased expression of the HO-1 isoform and higher production of CO decrease the production of reactive oxygen species (ROS) and suppress apoptosis (Pileggi et al., 2001; Li et al., 2016). We also assumed that HO-1/CO has the anti-apoptotic effect in oocytes. The increased expression of HO-1 during the in vitro aging of porcine oocytes, which we described, is likely induced by an increased production of ROS. High level of ROS is considered to be a major factor responsible for negative signs of aging in aged oocytes (Lord & Aitken, 2013). We also observed an increased expression of the HO-2 isoform. HO-2 is generally considered to be a constitutively expressed isoform responsible for a stable production of CO, whose level of expression does not respond to the effect of stress factors (Turkseven et al., 2007; Muñoz Sánchez & Chánez-Cárdenas, 2014). However, some authors have reported that in response to the stress factors the expression of HO-2 can also increase, which has cytoprotective effect (Kim et al., 2008; Ding et al., 2011). We suggest that both HO-1 and HO-2 can modulate ROS levels and suppress aging process in porcine oocytes.

The HO isoforms differed in their intracellular localization. It was found that the HO-1 was localized in meiotically mature and aged porcine oocytes predominantly in the area of chromosomes, and during the in vitro aging the HO-1 expression increased primarily in the cytoplasm. On the other hand, HO-2 was primarily localized in the cytoplasm. In general, HO-1 and HO-2 are considered to be proteins bound to the membrane of endoplasmic reticulum (Yoshida & Kikuchi, 1978; Ma et al., 2004), however, they may also be localized in other cellular compartments. HO-1 may be localized in caveolae, mitochondria and the nucleus (Dunn et al., 2014). Nuclear translocation of HO-1 requires cleavage of the membrane bound domain yielding ∼28 kDa protein fragment (Lin et al., 2007; Linnenbaum et al., 2012). The transfer of HO-1 into the nucleus takes place in the reaction to stress factors, and the nuclear HO-1 protects cells from the effect of oxidative stress (Lin et al., 2007). We assumed that this isoform may have similar function also in porcine oocytes. Meiotic maturation of oocytes under in vitro conditions is endangered by increased oxidative stress in comparison with in vivo conditions (Dvořáková et al., 2016). Because HO-1 is present in the perichromosomal area in matured oocytes, we assumed that the transfer of HO-1 could have already taken place during the meiotic maturation, due to possible oxidative stress.

We had proved that in porcine oocytes, the HO/CO contributes to sustaining their viability and affects the regulation of apoptosis. HO inhibition using Zn-PP IX leads to apoptosis in somatic cells (Hirai et al., 2007), which we also observed in our experiments. Porcine oocyte aging in the presence of the HO inhibitor worsened the effect of aging and increased the ratio of apoptotic oocytes. In the oocytes aged 24 h, the effect of the inhibitor was observable in the group cultivated with the highest concentration of the inhibitor (25 µM). In the oocyte aged 48 h, the effect of the inhibitor was significant in all concentrations; however the differences between the individual concentrations on proportion of apoptotic, parthenogenetically activated and intact oocytes were not significant. Given that the Zn-PP IX, particularly in higher concentrations, may also inhibit nitric oxide synthase (NOS) and soluble guanylyl cyclase (sGC) (Luo & Vincent, 1994; Grundemar & Ny, 1997), the possibility that this effect also appeared in our experiments cannot be excluded. However, Appleton et al. (1999) state that at the low concentrations of Zn-PP IX (up to 5 µM) there is selective inhibition of HO activity with minor effect on sGC and NOS. Therefore we assume that the non-selective inhibition could be seen only at the highest concentration (25 µM) used by us. Although the inhibition of sGC leads to the increased occurrence of negative signs of oocyte aging (Goud et al., 2005), the NOS inhibition, on the contrary, decreases the ratio of apoptotic oocytes (Nevoral et al., 2014). The increased production of NO through inducible-NOS (i-NOS) isoform in the cell contributes to the formation of oxidative stress (Wang et al., 2007b). It is therefore possible that while the inhibitor Zn-PP IX suppresses the protective effect of HO/CO, it also simultaneously decreases oxidative stress through iNOS inhibition.

This work is the first that proved the cytoprotective effect of CO on oocyte aging. As a source of CO we used CORM-2 and CORM-A1. CORM-2 and CORM-A1 both have distinct rates of CO release (Motterlini et al., 2002; Motterlini, Mann & Foresti, 2005). Both CO donors increased the ratio of intact oocytes and decreased the ratio of apoptotic oocytes during in vitro aging. In somatic cells, it was proved that HO/CO affects the viability of cells and apoptosis through the regulation of activity of pro-apoptotic and anti-apoptotic proteins. CO decreases the level of apoptosis, for example, through up-regulation of expression of the anti-apoptotic factor Bcl-2 and down-regulation of the activity of caspase-3 (Zhang et al., 2003a; Cepinskas et al., 2007). Caspase-3 activation also occurs during oocyte aging (Zhu et al., 2015; Zhu et al., 2016). As apparent from the results obtained, the CAS-3 attains highest activity in oocytes during the first day of aging, while over the course of the following days the activity of CAS-3 is significantly lower. Suppressing its activity through the CO donors affected the entire course of cultivation of the aging oocytes; but it was most significant, precisely during the first 24 h of aging. However, the fact that the effect also lasted through the following days of aging could be a consequence of the fact that CO has the ability of pre-conditioning, where a short exposure to CO subsequently leads to increased resistance of cells against stressors and thus to decreased level of apoptosis (Queiroga et al., 2012; Andreadou et al., 2015). Both CO donors suppress apoptosis even each of them has slightly different effect. The effect of CORM-2 on the ratio of apoptotic oocytes and the CAS-3 activity was not dose-dependent. However, in the case of CORM-A1, we observed the effect of different concentrations. We attribute it to the different kinetics of CO release from CORM-A1 and CORM-2 (Motterline & Foresti, 2017).

The mechanism through which CO affects CAS-3 activity is apparently complex, because CO may regulate the apoptotic pathway by multiple mechanisms. Beside the modulation of ROS levels, CO can modulate activity of c-jun terminal kinase (JNK), a member of the MAPK family. In somatic cells, CO decreases the activity of JNK (Morse et al., 2003) while in aging of porcine oocytes, it leads to the decrease of its activity and suppression of apoptosis (Sedmíková et al., 2013). Additionally, CO may suppress apoptosis through the modulation of the level of Ca2+. In somatic cells, CO may decrease the Ca2+ level (Lin & McGrath, 1988; Gende, 2004; Lim et al., 2005), while in aging oocytes the release of Ca2+ can trigger the cell death (Gordo et al., 2002; Zhu et al., 2016). Another molecular target for CO includes pro-apoptotic and anti-apoptotic factors. In somatic cells, CO increases the expression of the anti-apoptotic factor Bcl-2 and decreases the expression of the pro-apoptotic factors Bid and Bax (Zhang et al., 2003a; Zhang et al., 2003b; Wang et al., 2007a). It is possible that CO may also act in a similar manner in oocytes.

Conclusions

These experiments have shown that the CO donors suppress negative signs of aging in porcine oocytes and inhibit apoptosis through reduction of CAS-3 activity. For these reasons, it can be assumed that the HO/CO system is functional in porcine oocytes and that it contributes to sustaining the viability of oocytes and regulates the programmed cell death of oocytes.

Supplemental Information

Table S1 The effect of inactive CORM-2 (iCORM-2) or inactive CORM-A1 (iCORM-A1) on porcine oocytes during in vitro aging

Oocytes were cultivated to metaphase II and then exposed to in vitro aging in a modified M199 medium supplemented with iCORM-2 (100 µM) or iCORM-A1 (100 µM) for 24, 48 or 72 h. Control groups of oocytes were cultivated in modified M199 medium containing DMSO (in the case of iCORM-2) or distilled H2O (in the case of iCORM-A1). DMSO or distilled H2O were added in an equivalent volume as iCORM-2 or iCORM-A1 A Significant differences in the ratio of oocytes between control and iCORM groups during 24, 48 or 72 h separately are indicated with different superscripts (P < 0.05). MII, metaphase II (intact) oocytes; A, apoptotic oocytes; L, lytic oocytes; PA, parthenogenetically activated oocytes.

Click here for additional data file.

Data S1 Raw data for Fig. 4

Click here for additional data file.

Data S2 Raw data for Fig. 5

Click here for additional data file.

Data S3 Raw data for Fig. 6

Click here for additional data file.

Data S4 Raw data for Fig. 7

Click here for additional data file.

Data S5 Raw data for Fig. 11

Click here for additional data file.

Data S6 Raw data for Fig. 12

Click here for additional data file.

Additional Information and Declarations

Competing Interests

Author Contributions

The authors declare there are no competing interests.

David Němeček conceived and designed the experiments, performed the experiments, analyzed the data, wrote the paper, prepared figures and/or tables.

Markéta Dvořáková and Ivona Heroutová performed the experiments.

Eva Chmelíková performed the experiments, reviewed drafts of the paper.

Markéta Sedmíková conceived and designed the experiments, reviewed drafts of the paper.

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
