# Peer review of "Anti-apoptotic properties of carbon monoxide in porcine oocyte during in vitro aging"

_PeerJ, doi:10.7717/peerj.3876_

## Round 0.1 · original submission · Major Revisions

The Reviewers each provide a number of specific comments that need to be addressed before further consideration.

Reviewer 1 ·

Basic reporting

The work is well written and presented and the data are clear.

Insufficient citations of the literature. The authors should be more comprehensive of the literature as it relates to reviews cited (e.g. Motterlini) as well as some of the seminal papers showing the effects of HO-1/CO to mediate apoptosis (Brouard, Zuckerbraun, et al.).

The results are relevant to the hypotheses.

Experimental design

The experimental design is reasonable, but not terribly innovative given the available tools by which to study the effects of HO-1/CO on apoptosis. The authors use an outdated CORM (CO Releasing Molecule) and are missing important controls that include the metal core and the inactive compound.

The use of a phamacologics to block HO-1 should be complemented with siRNA validation given the off-target effects of Zn-PP.

The use of immunohistochemistry should be validated with PCR or Western blot data to show that there is true changes versus artifact of the staining and specificity of the antibodies.

From a scientific standpoint, none of the data is terribly novel given the enormous literature base. Adding a cell target for CO would strengthen the work and increase the enthusiasm that this will be an advance for the field.

Validity of the findings

Data is not terribly robust with the majority of findings dependent on immunostaining. Validation with qPCR or western blot is necessary. Given that both HO-1 and HO-2 are changing almost identically draws question as to whether the antibodies are specific. No controls are provided. HO-2 is typically constitutively expressed and if there is induction, this could be an important contribution.

Conclusions are reasonable, but again based on limited data sets that confirm a large literature base. Adding more detail on cell targets would be a stronger advance for the field.

Reviewer 2 ·

Basic reporting

This work studied the role of CO as an antiapoptotic molecule in porcine oocytes. The paper is easy to read and the basic ideas are clear. The english and references are appropiated. As metioned the structure is fine, but the some figures are dificult to understand and some results needs to be more robust.

Specific comments:

Page 7, line 38, please change "animal species" for "mammals"
Page 9, line 97, please change "carbon monoxide" for "CO"
Page 9, line 126, why above do you use ethanol and acetic acid and then changed to paraformaldehyde to fix the cells ?

In figure 2, change HO-1 for HO-2

Figures 4 and 5 are difficult to understand, can you show your results in a clearer way ?

In all the figures, please change the A, B, C nomenclature for more traditional *,**,*** for statitical significance.

Why in the text you add the mean values without error or standar deviation ?

Is the entire HO-1 localized in the nucleus or is a fragment of it, which could be recognized by the antibody ?

I have some concerns in relation with the HO inhibitor used in this work. In my undestanding (and as you commented in the discusion secction) the Zn-PP IX is a HO inhibitor but also inhibits other enzymes and receptors such as for example: interleukin-1 receptor, GCS, and all the NOS isoformos. Therefore, the results obtained could be subtantially improved if iRNA agaist HO is used.

Additionally, is suggested to use another CO donors with different structures in order to compare your results (e.g. CORM-A1). This because CORM-2 as well known produces CO, but also produce several other molecules, are you confident that any of these additional molecules are not doing anything in your model ?

Experimental design

The experimental design is fine, just minor comments, see above. The only concern is that the pharmacology can be improved (see comment above)

Validity of the findings

The results are novel, but they are not so robust, they need more work in terms of pharmacology and use of iRNA.

Conclusions are well stated and not to speculatives.

---

## Round 0.2 · Major Revisions

Each of the reviewers have provided focused, detailed, and helpful comments to complete the submission of your work. I hope you find them useful and I do believe you can complete them promptly and efficiently.

Reviewer 1 ·

Basic reporting

The responses of the authors were reasonable. unfortunately, the data in figures 3 and 10 are not complete. The authors need to show side by side the changes in HO-1 and caspase 3 cleavage. In figure 11, why is the dose response curve showing increased apoptosis with higher amounts of CORM at 24h?

Experimental design

reasonable except for the comments above about including comparators.

Validity of the findings

In light of the newly added data, it is difficult to see how the conclusions are being made and now need additional data.

Additional comments

Overall, the responses are adequate, but the data are somewhat incomplete (see above).

Reviewer 2 ·

Basic reporting

The manuscript have been improved. The authors asnwered to all my main concerns. However, still remains some point to be improved.

In the introduction between lines 72 and 88 is given the same (more or less) information compared to the paragraph between lines 89-97. Please fuse them in only one paragraph.

Line 238, change "they were" by we

In Results. Figure 2 must be the new figure 1.Therefore the sentece between lines 199 and 205 should be

"The objective of the experiment was to prove the presence of HO-1 and HO-2 proteins in meiotically mature porcine oocytes (MII) and oocytes exposed to in vitro aging for the period of 24, 48 and 72 hours. HO isoforms are present in porcine oocyte (FIG 1, ex figure 2), and their expression gradually increases during in vitro aging. In case of HO-1, the signal was predominant in the nucleus/perichromosomal area (Fig 2). On the contrary, in HO-2 the signal was primarily observed mainly in the oocyte cytoplasm (Fig. 3)."

In Discussion: if Zn-PP IX can inhibits the NOS...why you do not try in inhibit the NOS with L-name or other and see if the effect is similar or complementary to the CO-induced effect.

Experimental design

NA

Validity of the findings

NA

Additional comments

NA

---

## Round 0.3 · accepted · Accept

I believe that the revisions adequately address the concerns of the reviewers.